# Status and Prospects of Fluorescence *In Situ* Hybridization Automation—A Survey among Laboratory Directors and Their Designates

**DOI:** 10.3390/genes13112098

**Published:** 2022-11-11

**Authors:** Jun Gu, Zhenya Tang

**Affiliations:** 1Cytogenetic Technology Program, School of Health Professions, The University of Texas MD Anderson Cancer Center, Houston, TX 77030, USA; 2Department of Hematopathology, Division of Pathology and Laboratory Medicine, The University of Texas MD Anderson Cancer Center, Houston, TX 77030, USA

**Keywords:** FISH automation, cytogenetic laboratory testing, survey

## Abstract

As a clinical diagnostic technique, fluorescence *in situ* hybridization (FISH) is simple, reliable, cost-effective and widely applicable. Due to technology advances, automation systems are adapted in FISH in different ways, involving all and/or some of the following procedural steps: sample processing, probe distribution, hybridization, post-wash, result analysis and/or final report preparation. To better understand the status and prospective of FISH automation, a survey has been recently performed among Cytogenetic Laboratory Directors and/or their designated Laboratory Managers, Supervisors or certified Cytogenetic Technologists. We present here the preliminary analysis of this survey, to advocate more discussion about standardization of the FISH automation as well as implementation of FISH automation as part of educational programs for Cytogenetic Technologists.

## 1. Introduction

Fluorescence *in situ* hybridization (FISH) is simple, reliable and cost-effective. It is a major technology widely applied for clinical diagnosis, especially for hematologic malignancies, even in the era of next-generation sequencing (NGS). Attributable to the advances of technologies, automation systems are adapted in FISH sample processing and/or result analysis in clinical diagnostic laboratories. The only available FISH automation survey data were from a participant summary report (PSR) from College of American Pathologists (CAP) between 2017 and 2020 [1]. It indicated that only 15–20% laboratories were using automation in their FISH processing or analysis then. Currently, there are no CAP/American College of Medical Genetics and Genomics (ACMG) guidelines/recommendations on FISH automation. Preliminary data are needed to explore the possibility of standardizing FISH automation in sample preparation, hybridization, data analysis and result reporting. In order to assess how FISH automation is adapted in clinical cytogenetics laboratories, we conducted a survey during the summer of 2022. The survey was built on a Qualtrics Survey platform, and the link was distributed to the Cytogenetic Listserv, where most of the members are Cytogenetic laboratory Directors and/or their designated Managers, Supervisors or certified Cytogenetic Technologists. Although the results of this survey should be considered as preliminary, they could trigger more discussion about FISH automation standardization. The results from this survey could be used in a large-scale survey with better design. The primary goals of this survey are to identify FISH steps that are suitable for automation, in order to improve FISH result quality and comparability as well as the importance and urgency of implementing FISH automation, as part of educational programs for Cytogenetic Technologists.

## 2. Survey Design

The survey was limited to one submission from each laboratory in order to avoid a redundancy in the data. Submissions from similar IP addresses were reviewed, and only the one with the most complete data was included for analysis. First, the tests and services provided by a clinical laboratory were queried. Only laboratories currently providing FISH testing were requested to complete the survey regarding FISH automation status. The pre-signal analytic steps include automated sample processing for fixation, automated FISH probe denaturation, application to specimen and hybridization and automated post-wash with a programmable auto wash system. The signal analytic steps include automated image acquisition (whole slide imaging), automated identification of Region of Interest (ROI), automated cell segmentation with/without manual review and automated signal counting with/without manual review. The post-signal analytic steps include automated counting result calculation and automated report generation.

## 3. Survey Sample Population

In total, 45 surveys were returned, of which 7 incomplete surveys were excluded, resulting in 38 surveys qualifying for this study. Since some respondents left 1 or 2 questions unanswered in the surveys, data might not add up to a total of 38 in all questions.

## 4. Survey Results

Table 1 summarized all survey questions and responses. There were 29 (76.3%) respondents who stated that their laboratories perform FISH testing for both hematologic and/or other cancer samples along with constitutional samples. Eight laboratories (21.1%) perform FISH testing for hematologic and/or other cancer samples only. One laboratory (2.6%) provides FISH testing for constitutional samples only. Twenty-seven (71.1%) respondents stated that they perform chromosome analysis by karyotyping, 25 (65.8%) perform FISH testing for blood samples, 28 (73.7%) perform FISH testing for bone marrow samples, 27 (71.7%) perform FISH testing for tissue samples, 24 (63.2%) perform micro-array analysis and 15 (39.5%) perform next-generation sequencing.

Sixteen (42.1%) respondents stated that automation was used for FISH sample processing. Eleven (28.9%) respondents stated that automated FISH spot counting was applied in their laboratories. However, the automated FISH spot counting was applied for FFPE samples in only three (7.9%) laboratories participating in this survey. We designed questions for automation based on pre-signal analytic, signal analytic and post-signal analytic steps.

In terms of automation for pre-signal analytic steps, tissue processing for fixation was automated in 12 (31.6%) laboratories. FISH probe denaturation and hybridization were automated in 10 (26.3%) laboratories. The post-wash with a programmable auto wash system was administered in 12 (31.6%) laboratories. Automated identification of ROI was adapted in two (5.2%) laboratories only. For signal analytic steps, image acquisition was performed using a whole slide scanner in six (15.8%) laboratories. Two (5.3%) respondents stated that their laboratories use automated cell segmentation with/without manual review. Nine (23.7%) use automated signal counting with/without manual review.

The response distribution for post-signal analytic step automation showed that automated counting result calculation in 10 (26.3%) laboratories only and automated report generation in 5 (13.2%) laboratories only.

## 5. Discussion

CAP surveys include questions about FISH automation. The CAP PSR for FISH provided a very good starting point for exploring current FISH automation status. Data from 2017 to 2020 included as many as 272 laboratories performing clinical FISH tests. During the three-year period, FISH counting was performed manually in the majority of CAP-accredited laboratories and ranged between 188 (81.4%) and 233 (85.6%) among all laboratories surveyed. A fully automated FISH counting method was then adapted in 10 (3.7%) to 13 (3.8%) laboratories, and a certain level of automation for FISH counting was applied in the rest of the laboratories surveyed (28/11% to 35/13.3%) [1]. However, the PSR did not explore FISH automation applied in many other steps that could potentially be automated then. Despite a smaller sample size (N = 38), our survey attempted to investigate FISH automation more closely, with steps involving both counting and non-counting aspects. The participants of this survey represented traditional clinical cytogenetics laboratories performing both karyotyping and FISH analysis in the same specimen. There are opportunities to automate many or even all steps involving FISH analysis, such as specimen/slide preparation, probe denaturation and hybridization, post-wash, image acquisition, etc. All these could contribute to improved workflow efficiency and quality. Our survey indicated that more than two-fifths of the FISH laboratories have automated their sample processing regardless of the sample types; approximately one-third of the FISH laboratories have applied automated spot counting in their FISH testing. Concerns about the outcome could explain why there is no laboratory using a complete automated process at this time. Currently, a completely automated process without manual quality checks in multiple steps seems unrealistic. A partially automated process allows manual quality checks to ensure accuracy of outcome.

Early studies validated the FISH pretreatment automation for different sample types [2]. There were also multicenter studies exploring the feasibility of FISH automation in either wet lab or imaging on FFPE samples [3,4,5]. It seemed sample processing was more likely to be automated, especially in laboratories with high-volume testing. With many new automated sample processors being commercially available and validated, more laboratories will take advantage of technological advances to automate wet lab processes. According to our survey, the percentages of laboratories performing automated FISH counting are still low but have apparently increased (28.9% versus 13.3% in the CAP PSR) over time. Overall, FISH counting automation has trended upwards for the last five years, with the exception of FFPE FISH counting. FFPE FISH counting requires a high level of manual review that is not suitable for full automation now. Automated counting facilitates automated result calculation (26.3%) and automated report generation (13.2%). Many automated FISH counting software could export results to a spreadsheet. The Laboratory Information System (LIS) needs to authorize a built-in function to import numerical results to report templates in each facility.

Another goal of this survey is to identify the area of FISH automation for training purposes. The American Society for Clinical Pathology (ASCP) Board of Certification (BOC) Cytogenetics (CG) Exam currently does not require any training on FISH automation. The National Accreditation Agency for Clinical Laboratory Sciences (NAACLS) has an outdated Entry-Level Competencies for Cytogenetics Technologists, and it does not include any requirement for FISH automation. Clinical laboratories train their staff when acquiring new instruments from different vendors for FISH automation. Training programs for Cytogenetic/Medical Technologists might not have resources to keep up with new automation training. With more FISH automation instruments becoming available and validated, teaching courses could be updated or at least be structured based on pre-analytic, analytic and post-analytic automation, regardless of vendors and devices.

In summary, this survey has touched the basics of FISH automation, mainly utilized in traditional cytogenetic laboratories in the US. Due to the variety of diagnostic services/tests provided by them, the degree of FISH automation varies among the survey-participating laboratories. We believe new surveys with more detailed information such as validation of FISH automation performed, number of cells/signals counted for each test (e.g., suspension FISH versus tissue FISH), limit of detection of tests by FISH automation, etc., are necessary to be organized by certain scientific communities, such as CAP and/or ACMG.

## Figures and Tables

**Table 1 genes-13-02098-t001:** A summary of survey questions and responses to them.

What testing area does your lab cover?
Hematologic and /or cancer samples	8	21.1%
Constitutional samples	1	2.6%
Both	29	76.3%
Total	38	
Service(s) your lab provide
Chromosome analysis via karyotyping	27	71.7%
FISH-blood	25	65.8%
FISH-bone marrow	28	73.7%
FISH-tissue	27	71.7%
Micro-array analysis	24	63.2%
Next-generation sequencing	15	39.5%
Total	38	
Do you use automation for FISH sample processing?
Yes	16	42.1%
No	22	57.9%
Total	38	
Do you use automated FISH spot counting in your lab?
Yes	11	28.9%
No	27	71.1%
Total	38	
Do you use automated FISH spot counting for FFPE samples?
Yes	3	7.9
No	13	34.2
Not answered	22	57.9
Total	38	
Pre-signal Analytic Steps
Automated FISH probe denaturation and hybridization	10	26.3%
Automated post-wash with a programmable auto wash system	12	31.6%
Automated image acquisition (whole slide imaging)	6	15.8%
Automated tissue (sample?) processing for fixation	12	31.6%
Automated identification of ROI	2	5.3%
Total	38	
Automated cell segmentation with/without manual review	2	5.3%
Automated signal counting with/without manual review	9	23.7%
Total	38	
Post-signal Analytic Steps
Automated counting result calculation (ratio, % signal patterns)	10	26.3%
Automated report generation (pie chart, table)	5	13.2%
Total	38

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
