# Peer review of "Status and Prospects of Fluorescence In Situ Hybridization Automation—A Survey among Laboratory Directors and Their Designates"

_genes, 2022, doi:10.3390/genes13112098_

Round 1
Reviewer 1 Report
Line 28: Please define PSR.
Line 54: please define ROI.
Lines 49-56: The definitions in this study of pre-analytic, analytic, and post-analytic phases are different from those in CAP checklists that are used routinely by clinical laboratories. See below an example.
https://www.labmanager.com/insights/clia-compliance-for-pre-analytic-analytic-and-post-analytic-testing-phases-3323
For example, the CLIA/CAP defined steps in analytical phase are counted as pre-analytic steps in this study: “Pre-analytic steps include automated sample processing for fixation, automated FISH probe denaturation, application to specimen and hybridization, automated post-wash with a programmable auto wash system, automated image acquisition (whole slide imaging), and automated identification of ROI.”
This conflict of definition needs to be resolved to not cause confusions in clinical practice and in educational programs for technologists.
Author Response
Line 28: Please define PSR. Please see track changes on revised manuscript page 1, line 28. Line 54: please define ROI. Please see track changes on revised manuscript page 2, line 59. Lines 49-56: The definitions in this study of pre-analytic, analytic, and post-analytic phases are different from those in CAP checklists that are used routinely by clinical laboratories. See below an example. https://www.labmanager.com/insights/clia-compliance-for-pre-analytic-analytic-and-post-analytic-testing-phases-3323 For example, the CLIA/CAP defined steps in analytical phase are counted as pre-analytic steps in this study: “Pre-analytic steps include automated sample processing for fixation, automated FISH probe denaturation, application to specimen and hybridization, automated post-wash with a programmable auto wash system, automated image acquisition (whole slide imaging), and automated identification of ROI.” This conflict of definition needs to be resolved to not cause confusions in clinical practice and in educational programs for technologists. We thank reviewer 1 for catching this. Survey questions used were all analytic. The better way to categorize them is a). Pre-signal analytic; b). Signal analytic; and c). Post-signal analytic. Please see track changes on revised manuscript page 2, line 50-60; line 80-91; and table 1. Please see the attachment
Reviewer 2 Report
This study focuses on using automation in conventional FISH technique in different laboratories. The survey is well-designed and the outcome of the survey is significant. This reviewer has minor suggestions.
1. Authors may highlight the difference or similarity in the outcome of a complete automated process and a partially automated process.
2. Authors may add their comments in DISCUSSION regarding the frequent use of the manual approach than the automated approach
Author Response
This study focuses on using automation in conventional FISH technique in different laboratories. The survey is well-designed and the outcome of the survey is significant. This reviewer has minor suggestions.
- Authors may highlight the difference or similarity in the outcome of a complete automated process and a partially automated process.
We have revised and added comments on page 4, line 114 using track changes: “Concerns about the outcome could explain why there is no laboratory using a complete automated process at this time. Currently a complete automated process without manual quality check in multiple steps seems unrealistic. Partially automated process allows manual quality check to ensure accuracy of outcome.”
- Authors may add their comments in DISCUSSION regarding the frequent use of the manual approach than the automated approach.
We have added comments on page 4, line 124 using track changes: “With many new automated sample processor being commercially available and validated, more laboratories will take advantage of technology advance to automate wet lab process.”
We have added comments on page 4, line 130 using track changes: “FFPE FISH counting requires high level of manual review that is not suitable for full automation now.”
